# Advances in Ionospheric Space Weather by Using FORMOSAT-7/COSMIC-2 GNSS Radio Occultations

**Jann-Yenq Liu** [1,2,3,*] **, Chien-Hung Lin** [4,*] **, Panthalingal Krishnanunni Rajesh** [4] **, Chi-Yen Lin** [1,2] **, Fu-Yuan Chang** [1] **, I-Te Lee** [5] **, Tzu-Wei Fang** [6] **, Dominic Fuller-Rowell** [6,7] **and Shih-Ping Chen** [4]

1   Center for Astronautical Physics and Engineering, National Central University, Taoyuan City 320317, Taiwan; chiyen.lin@g.ncu.edu.tw (C.-Y.L.); fychang1228@gmail.com (F.-Y.C.)
2   Department of Space Science and Engineer, National Central University, Taoyuan City 320317, Taiwan
3   Center for Space and Remote Sensing Research, National Central University, Taoyuan City 320317, Taiwan
4   Department of Earth Sciences, National Cheng Kung University, Tainan City 70101, Taiwan; pkrgere@gmail.com (P.K.R.); z10802071@email.ncku.edu.tw (S.-P.C.)
5   Space Weather Operation Office, Central Weather Bureau, Taipei City 100006, Taiwan; itlee@cwb.gov.tw
6   NOAA Space Weather Prediction Center, Boulder, CO 80305, USA; tzu-wei.fang@noaa.gov (T.-W.F.); dominic.fuller-rowell@noaa.gov (D.F.-R.)
7   CIRES, University of Colorado Boulder, Boulder, CO 80309, USA
*   Correspondence: jyliu@jupiter.ss.ncu.edu.tw (J.-Y.L.); charles@mail.ncku.edu.tw (C.-H.L.)

**Abstract:** This paper provides an overview of the contributions of the space-based global navigation satellite system (GNSS) radio occultation (RO) measurements from the FORMOSAT-7/COSMIC2 (F7/C2) mission in advancing our understanding of ionospheric plasma physics in the purview of space weather. The global positioning system (GPS) occultation experiment (GOX) onboard FORMOSAT-3/COSMIC (F3/C), with more than four and half million ionospheric RO soundings during April 2006–May 2020, offered a unique three-dimensional (3D) perspective to examine the global electron density distribution and unravel the underlying physical processes. The current F7/C2 carries TGRS (Tri-GNSS radio occultation system) has tracked more than 4000 RO profiles within ±35° latitudes per day since 25 June 2019. Taking advantage of the larger number of low-latitude soundings, the F7/C2 TGRS observations were used here to examine the 3D electron density structures and electrodynamics of the equatorial ionization anomaly, plasma depletion bays, and four-peaked patterns, as well as the S4 index of GNSS signal scintillations in the equatorial and low-latitude ionosphere, which have been previously investigated by using F3/C measurements. The results demonstrated that the denser low-latitude soundings enable the construction of monthly global electron density maps as well the altitude-latitude profiles with higher spatial and temporal resolution windows, and revealed longitudinal and seasonal characteristics in greater detail. The enhanced F7/C2 RO observations were further applied by the Central Weather Bureau/Space Weather Operation Office (CWB/SWOO) in Taiwan and the National Oceanic and Atmospheric Administration/Space Weather Prediction Center (NOAA/SWPC) in the United States to specify the ionospheric conditions for issuing alerts and warnings for positioning, navigation, and communication customers. A brief description of the two models is also provided.

**Keywords:** FORMOSAT-7/COSMIC-2; GNSS RO; radio occultation; low-latitude ionosphere



## 1. Introduction

Radio occultation (RO) technique detects the changes to a radio signal when it passes through the atmosphere of a planet, and extracts information about the atmosphere based on the detected changes [1]. Though initially used for probing planetary atmospheres, the RO technique has been recognized as one of the effective diagnostic tools for remotely sensing the state of the atmosphere and ionosphere, especially as the global positioning system (GPS) signals were made accessible to the scientific community. The changes

in the relative positions between a GPS satellite and a low-Earth orbit (LEO) satellite allow us to determine the vertical profiles of temperature and water vapor pressure in the atmosphere as well as the electron density in the ionosphere. In 1995, the first low-Earth orbit satellite, MicroLab-1, receiving GPS signals demonstrated the application of limb sounding to probe the Earth's atmosphere and ionosphere [2–4]. Though the method has the potential of providing uninterrupted data across the globe, until the launch of the FORMOSAT-3 Constellation Observing System for Meteorology, Ionosphere, and Climate (FORMOSAT-3/COSMIC or F3/C), RO was limited to only a handful of attempts.

The F3/C mission, a joint Taiwan–United States (US) space program, with six identical micro-satellites placed into circular orbits with 72 deg inclination at ~800 km altitude, revolutionized ionospheric research, providing three-dimensional (3D) global electron density profiles, unaffected by measurement location or observing conditions [5]. The electron density profiles are retrieved from the GPS signals received by the GPS occultation experiment (GOX), the primary payload of F3/C. Interested readers may refer to Anthes et al. [6] for the details of deriving electron density profiles in the ionosphere as well as temperature, pressure, and water content profiles in the atmosphere by the RO technique based on the time delay and the bending angle of the GPS signals.

Launched in the beginning of the second quarter of 2006, the F3/C satellites offered unprecedented amounts of ionospheric soundings, yielding about 2000–3000 RO events per day, providing researchers worldwide with the opportunity to examine 3D ionospheric characteristics in greater detail, unlike what was possible before. The following years witnessed a large influx of quality research articles appearing in international journals, making use of the rare ionospheric sounding measurements from a unique space-based observing platform and portraying 3D electron density features and their temporal evolutions for the first time from actual measurements. There were several commendable efforts to validate the F3/C measurements by comparing with ionosonde and incoherent scatter radar observations [7–12], other in situ measurements [13,14], and model predictions [7,15–17], and to develop global models of ionospheric parameters based on F3/C measurements [15–19]. The results also led to several investigations to mitigate the impact of spherical symmetry assumption on the Abel inversion retrieval of F3/C RO electron density profiles [20–22]. Amidst such validation studies, the F3/C observations were applied to numerous research applications such as electron density responses to 27-day solar rotation [23], extracting plasmaspheric electron content [24], ionospheric irregularities from the fluctuations of signal-to-noise ratio (SNR) and phase data [25–34], retrieving topside scale heights [35] and equivalent ionospheric slab thickness [36,37], diurnal, seasonal, and longitudinal variations of electron density distribution [38–46], new aspects of middle-latitude trough [47] as well as mid-latitude summer nighttime anomalies [48–50], ionospheric perturbations by seismic activities, severe weather systems, solar eclipses, and geomagnetic disturbances [51–58], long-term ionospheric variations [59], etc.

Liu et al. [60] provided a detailed overview of such results, highlighting how the F3/C observations contributed to advancing the theoretical understanding of the evolution of the equatorial ionization anomaly (EIA), leading to the identification of new features such as plasma caves and plasma depletion bays, throwing light on the complex processes of coupling from the lower atmosphere that influences the electron density modulations in the ionosphere and new aspects of middle-latitude trough as well as mid-latitude summer nighttime anomalies. They also illustrated the application of F3/C measurements in examining ionospheric signatures induced by seismic and tsunami waves and earthquake precursors, as well as elaborated the global ionosphere specification (GIS) 3D electron density profiles constructed by using RO slant total electron content (TEC) measurements [61,62]. Note that, despite having 2000–3000 daily occultations, most of such investigations had to rely on average electron densities by combining observations over several days and within relatively larger longitude and latitude grids and time intervals.

After such dedicated service lasting about 14 years, far exceeding the expected mission lifetime, which saw more than 4.6 million RO profiles, the F3/C mission was officially

decommissioned in May 2020. A follow-up mission named FORMOSAT-7/COSMIC-2 (F7/C2) has already been launched on 25 June 2019, which also consists of a cluster of six satellites identical to that of the F3/C, circumnavigating the Earth with a period of about 97 min from low-inclination (24°) orbits at an altitude of ~550 km [63]. The Tri-GNSS radio occultation system (TGRS), the main mission payload of F7/C2, receives the refracted signals from GNSS (global navigation satellite system) satellites, including GPS, GLONASS, and Galileo systems, providing many more soundings than its predecessor. This results in, within ±35° latitudes, the daily ionospheric RO profiles of F7/C2 being about four times more than those of F3/C, which allows scientists to examine the equatorial and low-latitude ionosphere with higher temporal and spatial resolutions. The signal quality is also improved through ensuring higher SNR by employing digital beamforming. Instead of the patched antenna arrays of F3/C, the F7/C2 uses advanced phase antenna arrays produced using 3D printing technology for the first time in space applications. The F7/C2 configuration provides approximately 4000 occultations per day, with unprecedented observations facilitating a better understanding of the structures and electrodynamics of the equatorial and low-latitude ionosphere, as well as opening a new chapter for the ionospheric weather forecast.

Lin et al. [64] performed the early validation of the F7/C2 electron density profiles and GIS by comparing with manually scaled F-region peak density and peak height from the Digisonde network, yielding correlations in the range 0.86–0.9. Furthermore, Lee et al. [65] carried out a detailed validation and investigation of the retrieved profiles, evaluating the reliability of the measurements and their suitability for ionospheric research and applications. The early orbital period of F7/C2, when the satellites were clustered together, provided information about ionosphere variations in finer temporal and spatial scales [66]. The F7/C2 measurements were used to investigate the role of atmosphere–ionosphere coupling in contributing to the day-to-day ionospheric variability [67], and in particular, to examine the temporal characteristics and 3D features of the electron density modulations during sudden stratosphere warming (SSW) [68]. The observations have also been used to investigate global ionosphere responses during minor magnetic storms [64,69]. In addition, the S4 scintillation index derived by using the F7/C2 measurements has been shown to be a promising data product for near real-time monitoring of equatorial plasma irregularities [70]. Such continuing inflow of quality research articles underscores the importance of global GNSS RO measurements in space weather monitoring and research.

This paper extends the work of Liu et al. [60] by examining the potential of F7/C2 in expanding our understanding of the various ionospheric features revealed by F3/C. Liu et al. [60] overviewed the F3/C results and pointed to the prospects with F7/C2. The focus of this paper was to give the readers an overview of how these prospects are realized, highlighting the added advantages of the new F7/C2 observations. Thus, the major achievements in the ionospheric research over the last decade by using F3/C observations were briefly re-constructed based on the F7/C2 soundings wherever possible. Section 2 demonstrates the advantage of the F7/C2 configuration by comparing the distributions of RO soundings with that of the F3/C. Section 3 describes the major achievements in understanding the low-latitude electron density distribution. The advantage of F7/C2 as a space-based platform for near-real-time global plasma bubble monitoring is elaborated on in Section 4. Finally, the role of the F7/C2 observations in achieving realistic space weather nowcast and forecast is described in Section 5 based on the numerical models developed at the Space Weather Operational Office at Central Weather Bureau (CWB/SWOO) of Taiwan and at the Space Weather Prediction Center at National Oceanic and Atmospheric Administration (NOAA/SWPC).

## 2. Intense GNSS RO Soundings of the Ionosphere

The scientific community, especially the atmospheric and ionospheric researchers, were widely anticipating the launch of F3/C because of the unique observations that, for the first time, offered 3D profiles of meteorological parameters as well as the ionospheric electron

density spanning across the globe. Figure 1a provides a comparison of the distributions of the daily occultations from F3/C and those from F7/C2 until 1 March 2022. During the initial 1-year period after the launch, the F3/C averaged about 2500 occultations per day, which occasionally crossed 3000. Note that during this period, the satellites were mostly clustered together, and undergoing various orbital maneuvers to deploy each satellite to their final obits, approximately 60° apart. Since then, until the middle of 2010, the average occultations were about 1500, which afterward dropped to about 1000–1200 and remained mostly so until the end of 2015. The last 5 years of the F3/C saw the occultations falling to much smaller numbers, with a couple of satellites being forced to terminate operations due to technical problems.

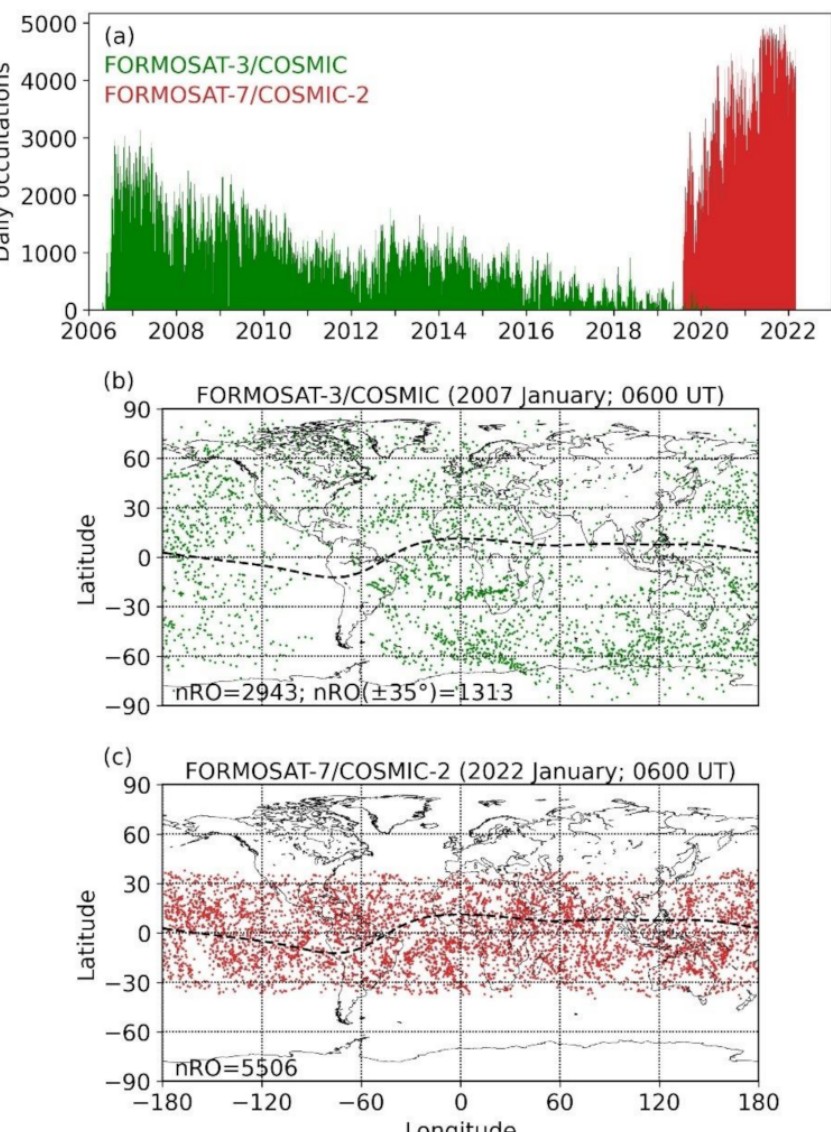

**Figure 1.** Comparison of F3/C and F7/C2 occultations and distribution of tangent locations. (**a**) All available occultations from F3/C mission from 2006–2020 (green) and the daily occultations from F7/C2 until 1 March 2022 (red). The number of occultations on a given day denotes the total number of retrieved electron density profiles (ionPrf). (**b**) The $NmF_2$ locations (green dots) from F3/C observations during January 2007 at 0600 UT. (**c**) Same as (**b**), but from F7/C2 during January 2022. The dashed curve denotes the magnetic equator. "nRO" stands for number of occultations, and "nRO ($\pm$35°)" in panel (**b**) represents the number of occultations within $\pm$35° latitudes.

In contrast, the number of occultations from the F7/C2 mission gradually increased after the launch, where the occasional intervals of reduced soundings corresponded to the phases when the satellites were being moved down from their initial parking orbits at about 700 km to their final mission altitudes at about 550 km altitude. The number of occultations gradually increased from the initial range of 1000–2000, reaching 3000–4000 daily occultations from early 2020, which have appeared to remain consistent at around 4000–4500 since the second quarter of 2021. Comparing the performances, except for the early months of the launch, F7/C2 achieves more than two times the number of occultations provided by F3/C.

Clearly, the F7/C2, with its Tri-G capability, outperforms its predecessor in terms of the number of daily soundings, which is welcome news to the community. It is also worth comparing the spatial distributions of the occultations. In Figure 1b, the month of January 2007 was selected to examine the RO locations for F3/C, which falls within the period of its best performance (Figure 1a). For F7/C2, a corresponding 1-month period in 2022 is plotted in Figure 1c. The figure shows the geo-locations of the F-peak electron density ($NmF_2$) during a 1 h period at 0600 UT. Though the spatial distribution of the occultation locations is comparatively sparse, the F3/C is advantageous when it comes to global coverage, scanning both high- and low-latitudes alike. Owing to the low-inclination orbit, the F7/C2 occultations are mostly within about $\pm35°$ latitudes, which effectively enhances the spatial resolution. The figure demonstrates that owing to the denser sampling of the low-latitude regions, the F7/C2 could unravel finer details of the thermosphere–ionosphere dynamics when compared to F3/C.

It can be further seen from Figure 1b that, though F3/C provided approximately 3000 occultations during the plotted period, only ~1300 soundings out of that fell within $\pm35°$ latitudes. On the other hand, F7/C2 sampled the same region during the same period with more than four times the corresponding number of F3/C occultations (Figure 1c). Note that the locations marked in Figure 1b,c correspond to the F-peak altitude (hmF2), and the actual latitude coverage of F7/C2 may be slightly larger when considering the tangent locations of the entire profile for each occultation event. This paper made use of these additional available measurements to re-examine the previously reported major results by F3/C.

## 3. Equatorial and Low-Latitude Ionosphere

This section provides an overview of the prominent features of equatorial and low-latitude ionosphere consisting of the evolution of EIA [38], plasma cave structures [39], longitudinal modulation of EIA by non-migrating tide of wave number four (wavenumber-4) [41,42], and plasma depletion bays [40] by making use of the enhanced Tri-G measurements from the F7/C2 constellation. This denser sampling allows scientists to depict the 3D structures and features in the equatorial and low-latitude ionosphere and thus examine the detailed temporal features of the associated electrodynamic processes.

### 3.1. Equatorial Ionization Anomaly

One of the fascinating features in the distribution of ionospheric electron density is the equatorial ionization anomaly, manifesting as two enhanced peaks (crests) of electron density surrounding a narrow belt of lower values of ionization (trough) roughly centered over the magnetic equator during the daytime [71,72]. Being the locus of the largest electron density, the EIA crests often attract the attention of the ionospheric community, which offers a crucial vantage point to investigate the equatorial and low-latitude ionospheric electrodynamics dominated by the equatorial plasma fountain and neutral wind [73,74]. These dynamic processes in the formation of EIA are susceptible to energy input from high latitudes [75–78], waves, and tides that propagate from the lower atmosphere [79–83], making the variabilities in the EIA electron density an effective sensor to monitor coupling from other regions.

Figure 2 depicts the horizontal and vertical distributions of the low-latitude electron density by combining 1-month F7/C2 measurements during a 1 h period centered at

1200 UT. The figures, adopting a similar format used by Lin et al. [38], highlight the potential of F7/C2 in providing the same information by combining only about a quarter of the time duration of observations that was required for the F3/C. The figure displays the monthly mean distributions of electron density in March, June, September, and December of 2021, revealing the seasonal characteristics. The largest electron densities occurred at the EIA latitudes over 250–300 km altitudes, spread across the longitudes of 60° W to 120° E. Notable seasonal differences existed in the low-latitude electron density, which was most pronounced in March and December with June yielding the lowest densities amongst the four months plotted. In the equinox months of March and September, the EIA peaks were more-or-less symmetric with respect to the magnetic equator. Though the overall distributions were mostly identical in the equinoxes, the electron density in March appeared slightly larger than that in September, revealing equinoctial asymmetry. Further, in September, there was an enhanced region of electron density over the west African and Asian longitudes of 60°–120° E in the Northern Hemisphere, causing a substantial deviation from the equinox distribution and resembling northern summer conditions.

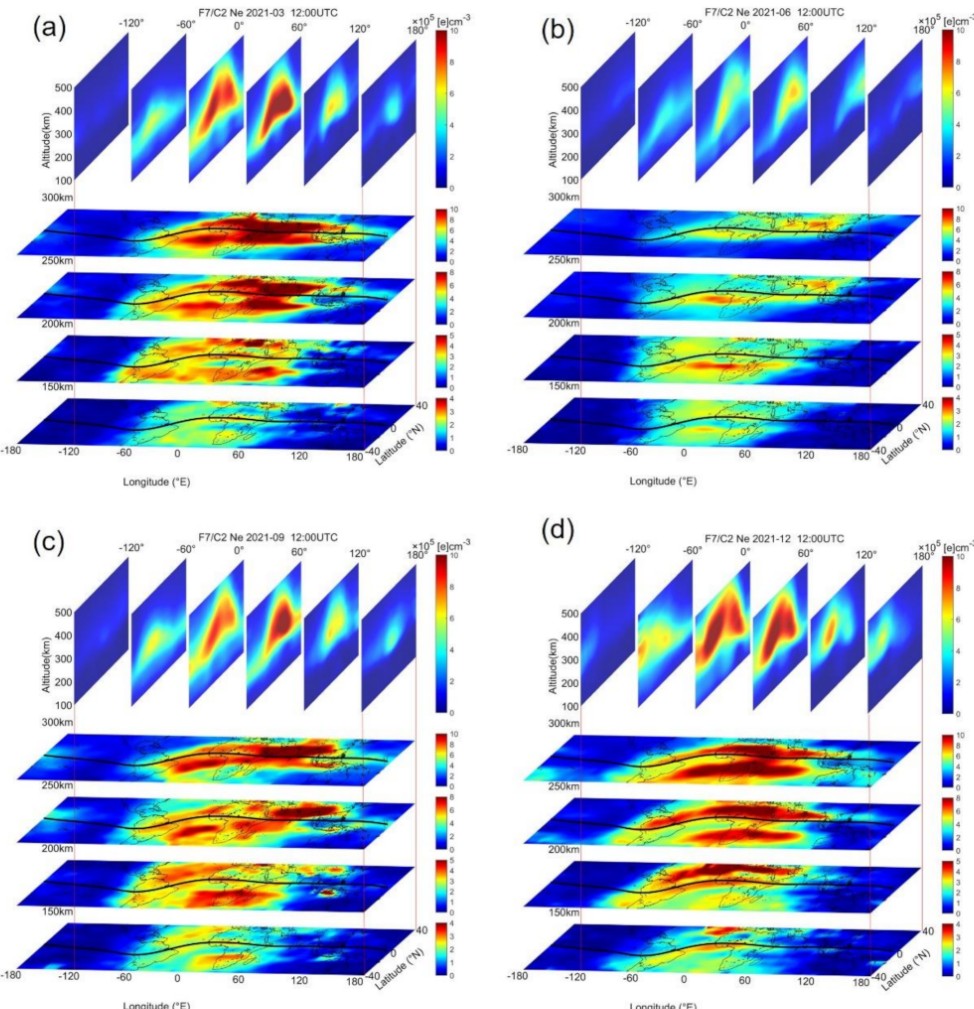

**Figure 2.** The 3D F7/C2 electron density structure at 12:00 UTC time in (**a**) March, (**b**) June, (**c**) September, and (**d**) December of 2021. The horizontal maps reveal well-developed EIA bands, being pronounced around 250–300 km altitudes over 60° W–120° E longitudes. The plasma caves appeared below 250 km in 0° E and 60° E longitude slices in the top panel. The median values of all the observations for each month at the selected hour are arranged in grids of 2.5° × 2.5° longitude–latitude, and 5 km altitude.

Note that equinox asymmetry in the electron density distribution has been reported with larger F-region electron density during March equinox compared to September equinox, which is attributed to asymmetries in neutral wind, temperature, and vertical plasma drifts [84,85]. Further, Ren et al. [86] examined the role of DE3 (diurnal eastward propagating wavenumber-3) tides in affecting the vertical ExB drifts and contributing to the asymmetry and suggested the asymmetry to be pronounced during low solar activities. Mehmet [87] reported the north-south asymmetry in the electron density distribution in equinoxes over African and west Asian longitudes. Balan et al. [43] examined the equinoctial north-south asymmetry over different longitudes and showed that the asymmetry was pronounced over the longitudes where the displacement of the geomagnetic equator from the geographic equator is larger.

In addition to such equinox asymmetry, the figure also shows a large difference in electron density during solstices, with a much stronger electron density in December than in June. This is attributed to the December anomaly or the annual anomaly [88] with more electron density occurring in the December solstice than in the June solstice. The hemispheric asymmetry or winter anomaly of the EIA peaks in the solstice months is also evident in the figure. The vertical slices in the top panels illustrate the artificial plasma cave feature, first shown by Liu et al. [39] by using F3/C observations. Interested readers may refer to Liu et al. [60] and the references therein for a detailed discussion of the plasma cave feature and its significance.

The illustrations given in Figure 2 and the features discussed show the importance of having more dense observations to unravel finer details of the ionospheric electron density distribution. A movie constructed by using such plots for all the 12 months in 2021 is provided in the Supplementary Materials (Movie S1), demonstrating the seasonal transition of the EIA development. The monthly electron density maps reveal further seasonal characteristics of electron density. For example, larger electron densities occurred in October, whereas earlier average plots combining 60 or 90 days of measurements would not be able to capture such seasonal transitions.

Since the F7/C2 measurements are limited to the low-latitude regions, we further focused on the diurnal evolution of EIA crests in Figure 3a–d, which shows the development of EIA crests at 120° E longitude in the solstice months of June and December, as well as the equinoctial months of March and September. June and December months reveal the general EIA morphology discussed by Lin et al. [38], with the electron density first appearing in the summer hemisphere followed by the characteristic winter anomaly showing enhanced EIA crests in the winter hemisphere in the morning sector, which gradually reverses in the afternoon period, yielding overall larger densities in the summer hemisphere. Tsai et al. [89] explained the combined theory of trans-equatorial neutral wind, subsolar point, and auroral equatorward wind in summer and winter solstices, producing such seasonal variations of the EIA crests.

In the equinox months, there was a noticeable difference in the EIA development between March and September months. In March, the electron density started to enhance over the magnetic equator, and after 1000 LT, the northern EIA crest began to develop. The southern crest also built up by around 1200 LT, with both the crests enhancing symmetrically in the subsequent hours, exhibiting a typical equinox behavior. However, in September, the EIA development was somewhat similar to solstice conditions, with the Northern Hemisphere having much stronger density than in the Southern Hemisphere. Such strong hemispheric asymmetry is unusual in the September equinox period, which suggests a possible delay effect. The detailed results showed that the EIA development in October was similar to that in March (Movie S2 in the Supplementary Materials).

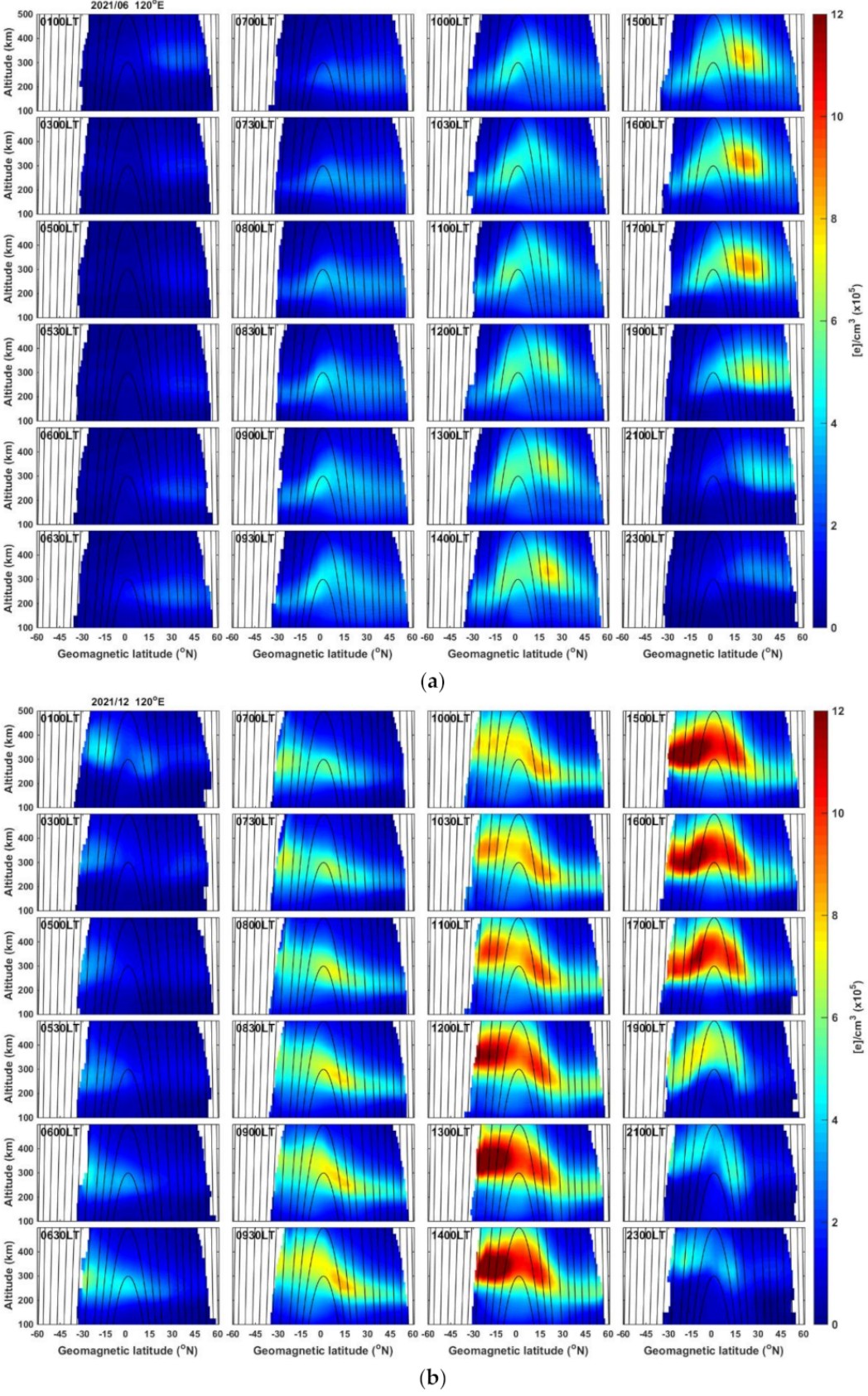

(**a**)

(**b**)

**Figure 3.** *Cont.*

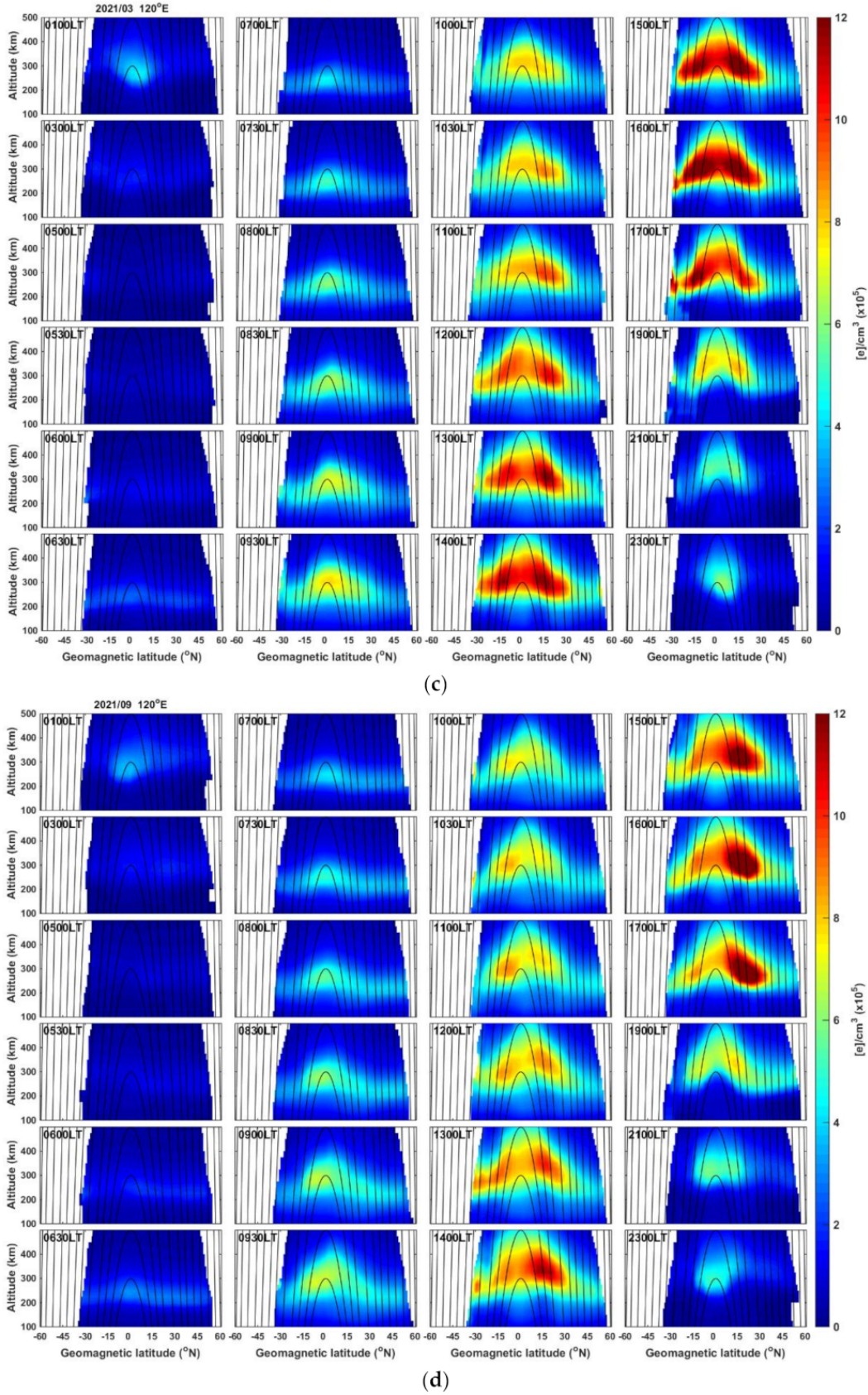

**Figure 3.** (**a**) Local time evolutions of the ionospheric electron density structure at the mid- and low-latitude regions at 120° E longitude sector from 0100 LT to 2300 LT in June 2021. The binning method was same as that for Figure 2. Gray curves represent the magnetic field lines. (**b**) Same as (**a**), but for December 2021. (**c**) Same as (**a**), but for March 2021. (**d**) Same as (**a**), but for September 2021.

On the other hand, there was a clear seasonal asymmetry in the morning hours where the electron density in the winter hemisphere generally was greater than that in the summer hemisphere in June and December. The electron density in December was much larger than in June, which may partly be due to the December anomaly [88]. However, again, the detailed results (Movie S2) showed that instead of December, the electron density in January was more similar to that in June, but with the opposite latitudinal distribution. Meanwhile, the large enhancement of the electron density in December 2021 might also be due to the increase in solar activity from December 2020 to December 2021, with the sunspot number (10.7 cm solar flux, F10.7) increasing from 23 to 67 (87 to 102 sfu). Nevertheless, these high temporal and spatial resolutions of F7/C2 RO observations have significantly advanced the understanding of the electrodynamics in the equatorial and low-latitude ionosphere.

In addition to the seasonal effect, we also examined the response of EIA evolution with varying magnetic declination. Figure 4 illustrates the time evolution of EIA over various altitude–latitude cuts along the longitudes of 165° W (declination angle 9.7°), 70° W (declination of −9.7°), and 50° W (−18.5°) in December 2021. In Figure 4a, the EIA development was more similar to equinox conditions, whereas a strong hemispheric asymmetry was expected for the December month, similar to that in Figure 4b,c. Among these three longitudes, the geomagnetic and geographic equators almost overlapped over 165° W and 50° W, but there was a large offset over 70° W. The mostly symmetrical EIA development over 165° W indicates the major role of the equatorial plasma fountain, with minimal contribution by trans-equatorial winds. However, over 70° W, neutral wind yielded an extremely strong north-south asymmetry. These variations agree with previous studies showing that neutral wind is more effective in re-organizing plasma distribution when the separation between geomagnetic and geographic equators is larger, with a much smaller role by the declination angle [43]. However, over 50° W, which is also similar to 165° W with large declination and overlapping equators, the neutral wind is still effective in modifying the latitude electron density structure. This hemispheric pattern at 50° W could not be well-explained by Balan et al. [43] and suggests that declination angle effects are also important. Possible longitudinal variations in neutral wind can also yield such differences. Meanwhile, the longitudinal features of wavenumber-4 [41,42,86] might also result in discrepancy between current and previous studies. Nevertheless, advances in F7/C2 RO observations should shed more light on uncovering the declination effects.

*3.2. Wavenumber Four and Plasma Depletion Bay*

In addition to the morphological characteristics, the longitudinal differences in the distribution of EIA electron density have been more intensely examined ever since the demonstration of the existence of the wavenumber-4 pattern in airglow images [90,91]. The observations demonstrate how the dynamic processes occurring at low altitudes can significantly modulate F-region electron density, providing indisputable evidence for the link between terrestrial weather and ionospheric variations. The absorption of solar radiation by water vapor and ozone drives sun-synchronous atmospheric tides [80–82], and latent heat release associated with deep tropical convection incites non-migrating tidal variations [83,92–94]. These dynamic variations in the lower atmosphere in the form of waves and tides propagate upward, where they grow in amplitude and eventually dissipate in the mesosphere–lower thermosphere (MLT) region around 80–150 km, depositing their energy and momentum and modifying the background temperature and wind [92,95]. Such tidal interactions modify the E-region dynamo electric fields that drive the equatorial plasma fountain, thus imparting the source variations all the way to F-region altitudes [96,97].

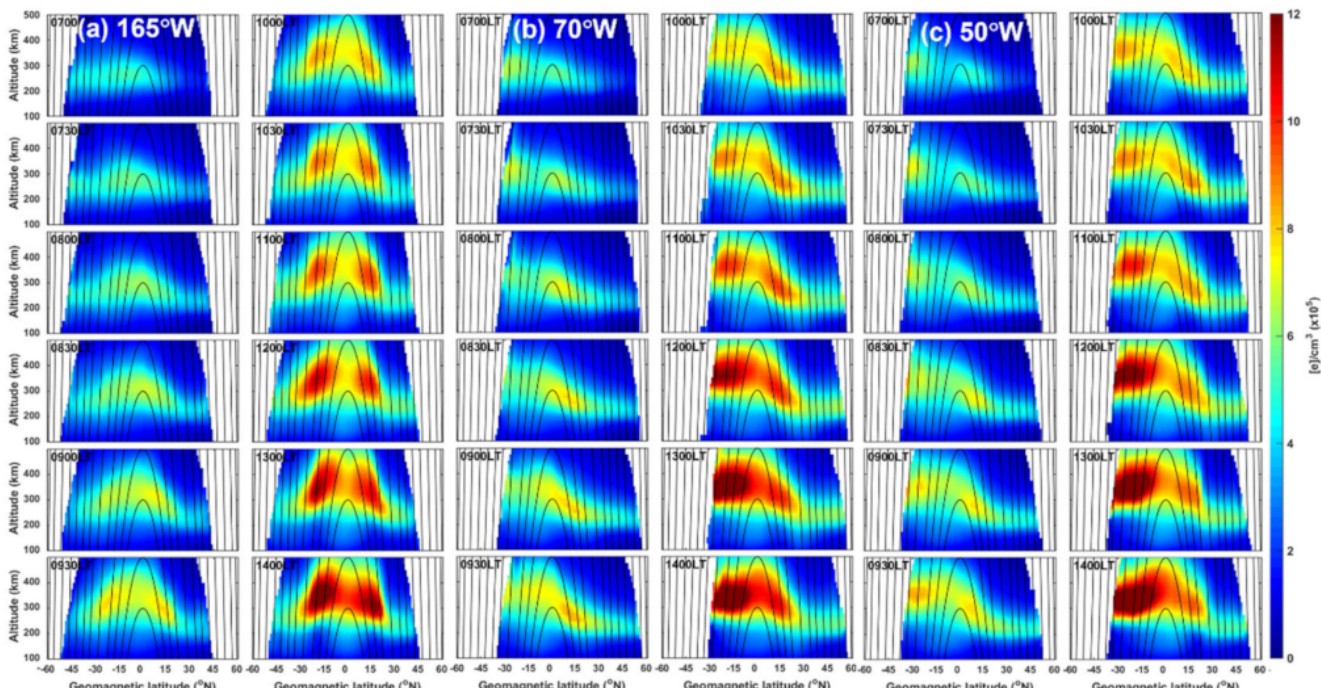

**Figure 4.** Local time evolutions of the ionospheric electron density structure at the mid- and low-latitude regions during 0700 LT - 1400 LT over (**a**) 165° W, (**b**) 70° W, and (**c**) 50° W longitude sectors in December 2021. The binning method was same as that for Figure 2. Gray curves represent the magnetic field lines.

　　　　The potential of F3/C RO observations was fully exploited to analyze the vertical structures and local time evolution of the wavenumber-4 pattern [41,42,97], and to identify the contributions of non-migrating tides and stationary planetary waves in modulating the low-latitude plasma distribution by modulating the ionosphere dynamo [98–101]. Figure 5 (left columns) illustrates the monthly variations of the longitudinal wave pattern at 1500 LT in 2020 at an altitude of 400 km. The modulation of the EIA predominantly displayed a wavenumber-3 pattern during January–June, which then evolved to the wavenumber-4 characteristic until about October, before returning to wavenumber-3 again. Thus, for the local-time period plotted in the figure, the overall EIA pattern throughout the year was dominated by a wavenumber-3 modulation. This monthly pattern generally agrees with seasonal variations of non-migrating oscillations extracted from F3/C measurements, with wavenumber-4 constituents yielding larger amplitudes around September equinox and wavenumber-3 amplitudes being relatively stronger in the solstice months [99,100]. The seasonal and inter-annual variability in the relative amplitudes of the migrating oscillations in electron density may arise from several factors such as source variations, changes of the mean flow in the MLT region affecting wave dissipation, variations of the zonal mean and diurnal components of ionospheric conductivity or neutral wind that alias with the non-migrating tides, producing child wave modulations, as well as non-linear interactions of migrating tides with local-time variations in the neutral atmosphere, yielding planetary-scale signatures [100].

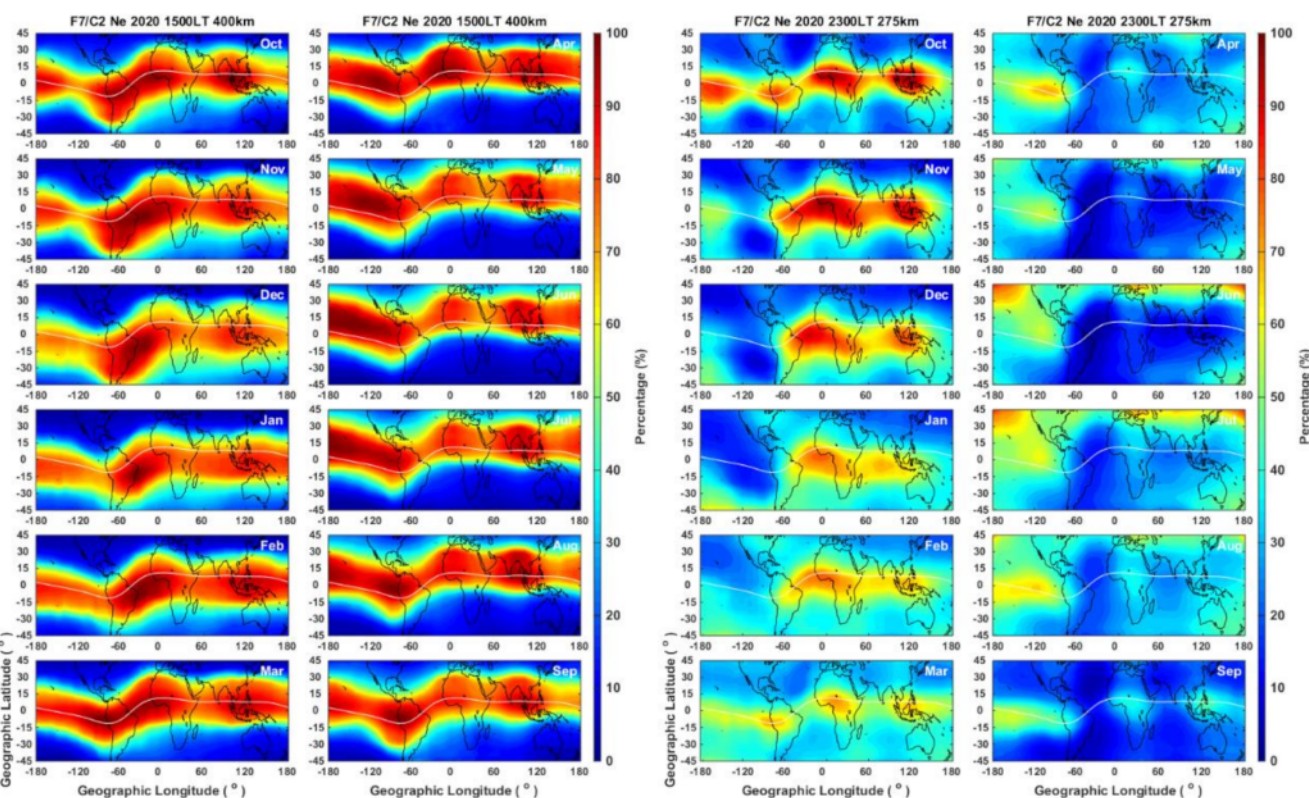

**Figure 5.** Monthly plots of the F7/C2 electron density at 400 km, 1500 LT (**left two columns**) and 275 km, 2300 LT (**right two columns**) in 2020.

Most of such earlier studies using F3/C measurements required a larger temporal window (30–90 days over a 2 h interval) and/or integrating the measurements over a given altitude range. The advantage of F7/C2 is the capability to reproduce similar results by combining observations within shorter time intervals, thus offering the opportunity for further investigations to understand variabilities in the factors that modulate the EIA electron density distribution. Though most of the discussion of the low-latitude electron density variations focuses on the influence of E-region dynamo, variations of neutral composition as well as F-region meridional winds by non-migrating tides can also contribute to the EIA modulations [102]. It is expected that the denser RO observations by F7/C2 may play a major role in further exploring the different nature of the forcings and help to gain a better insight into the complex wave interactions in the upper atmosphere and how such different factors may contribute to the seasonal and inter-annual variations as well as hemispheric asymmetry of the wavenumber-4 modulation. In addition, the GIS electron density profiles constructed by using F7/C2 observations offer the unique opportunity to explore day-to-day variability of such longitudinal modulations [67] and a more detailed understanding of the electron density modulations forced from the variations below, such as the ionosphere EIA response to SSW events [68].

Apart from the wavenumber-4 longitudinal modulation of EIA crests that is widely investigated, a new type of electron density structure was reported by Chang et al. [40], manifesting as a large-scale depletion of electron density over certain longitude regions. This new feature, named plasma depletion bay (PDB), was identified by examining the 3D global electron density maps during nighttime by combining about 60 days of F3/C observations at fixed local-time frames. The results showed a single and intense PDB originating from the Northern Hemisphere (north PDB) over 75–135° W during the months of October–March and three such PDB features originating from the Southern Hemisphere (south PDBs) over the North Atlantic (60° W to 30° E), Indian Ocean (45–110° E), and Southeast Asia (120–170° E) longitudes during April–September. Unlike the wavenumber-4

modulations, the PDB structure pertained to nighttime plasma dynamics, being prominent at around 2300 LT, and their generation was attributed to the field-aligned plasma transport by the magnetic meridional wind from summer to winter hemisphere [40].

The global maps at 2300 LT in the right columns in Figure 5, constructed by using F7/C2 observations at 275 km altitude, illustrate the evolution of the north PDB and the south PDBs. When compared to the results using F3/C [40], the F7/C2 electron density offers more distinct monthly variations, with the PDB feature appearing pronounced in the solstice months and remnant wavenumber-4 modulation persisting in the equinox months. The PDB morphology, especially that of the north PDB, with the depleted plasma regions aligned normal to the magnetic equator, ascertains the peculiar role of the field-aligned component of the summer-to-winter neutral wind in removing the ionization in both the hemispheres alike. Additionally, note that the PDBs originate in the summer hemisphere over the longitudes where the separation between geomagnetic and geographic equators is largest. In other words, the PDBs occur near the longitudes where the magnetic equator is more offset to the summer hemisphere. Over such longitudes, the magnetic meridional wind is more effective in the inter-hemisphere plasma transport along the field lines [43].

Such features of the low-latitude ionosphere, especially over the EIA region, highlight how the GNSS RO measurements continue to play an important role in advancing the current understanding of the low-latitude plasma dynamics and atmosphere–ionosphere coupling. The comparison of the major findings by using the F3/C measurements provided here and re-producing similar results based on the F7/C2 data demonstrate how the increased observations help to unravel additional details of the low-latitude variations. The advantage of the current F7/C2 sampling is the ability to construct global electron density maps at every hour in each month, offering more detailed insights into diurnal and seasonal features in the electron density variations.

## 4. Near-Real-Time Global Plasma Bubble Monitoring

In addition to the opportunity to better explore the characteristics of low-latitude electron density distribution as described above, the increased F7/C2 occultation soundings also provide more closely sampled measurements that can be effectively utilized in monitoring occurrence and distribution of ionospheric irregularities [103–105]. This section describes the potential of applying the F7/C2 soundings as an effective tool for near-real-time global plasma bubble monitoring.

The scintillation index S4 (S4-index) derived from the SNR of receiving GNSS L1-band (1.575 GHz) C/A code in the F3/C scnLv1 data product has been utilized to observe global plasma irregularity occurrence in the ionosphere [106,107]. For ground-based and aviation communication applications, Liu et al. [103] developed a method for integrating the scattered distributed F3/C S4max in the ionosphere vertically to the ground, while Chen et al. [104] constructed an empirical global model, the F3CGS4 (F3/C Global S4) model, to calculate the climatological median S4-index for a designated year and day of the year, local time, location, and solar activity F10.7 index. Chen et al. [108] further improved the F3CGS4 model by using the whole S4-index profile data and constructing the global probability model for the S4-index of L-band scintillations. The dense F7/C2 radio occultation scintillation provides a good chance to observe equatorial and low-latitude ionospheric GNSS L-band S4 scintillations.

*Equatorial and Low-Latitude Ionospheric GNSS L-Band S4 Scintillation*

The F7/C2 radio occultation scintillation is recorded in both 1 and 10 s temporal resolutions (5–10 km and 10–15 km in altitude, respectively). It is calculated by receiving the L1-band (1.5 GHz) CA code of the GPS and GLONASS satellite systems. The F7/C2 provides more than 5000 RO S4-index vertical profiles per day, uniformly observing the mid- and low-latitude (between 40° N to 40° S) ionosphere (50 to 550 km in altitude) of the globe every hour. The RO S4-index is considered as one of the indicators of the equatorial

plasma bubble (EPB) in the nighttime ionosphere. Therefore, by utilizing the F7/C2 RO S4 data, it is possible to monitor global EPB near-real-time occurrence.

Figure 6 displays the distribution of nighttime intense scintillation (S4-index > 0.5) on 1 January (DOY001) 2022. The occurrence of S4 > 0.5 appeared dominant in between 0° E and 60° W. The occurrence pattern in the longitude–altitude intersection of the geomagnetic equator (the rear diagram) shows both west and east-tilted stripes, indicating the co-existence of C-shape and reverse C-shape EPBs within the longitude sector. In the latitude–altitude intersection of 20° W longitude (the side diagram), the asymmetry double-peak structure at 15° N and 15° S displays the conjugacy of the EPB across the geomagnetic equator. The EPB conjugate pattern can also be seen in the horizontal longitude–latitude diagram of 300 km altitude (the bottom diagram).

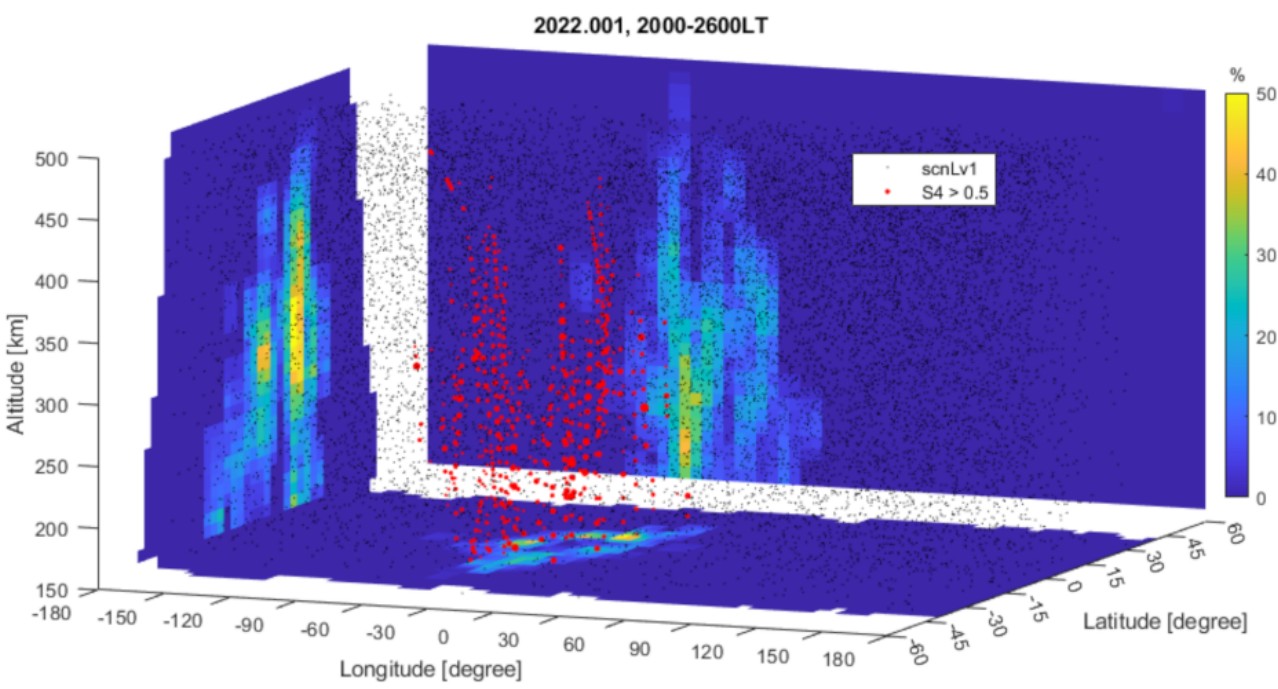

**Figure 6.** The occurrence of intense S4-index within one night. The color maps indicate the occurrence of intense S4 (red solid dots) in latitudinal (rear diagram), longitudinal (side diagram), and altitudinal intersections (bottom diagram) during 2000 LT to 0200 LT of 2022.001 to 2022.002.

The F7/C2 S4-index is employed to construct a global EPB occurrence database, which monitors the day-to-day variability of the initiating hour and the total duration time of this particular phenomenon. Figure 7 displays the initiating time and the overall duration of the EPB in low latitudes (between 30° N and 30° S) since the launch of the F7/C2. The results showed that EPBs frequently appeared from 20° W to 80° W during September and March. The EPBs became scattered worldwide in early August until the end of September, frequently occurred over 0–60° W during September–March, started scattering worldwide from February to April, and yielded few signatures around 0° and ±180° longitude during March–September. EPBs mainly appeared between 1800 and 2300 LT. It was found that the EPBs tended to appear earlier and yield a longer duration during the higher solar activity period such as in 2021. Nevertheless, Figure 7 demonstrates that the F7/C2 RO observations can be used to monitor global EPBs in near-real-time.

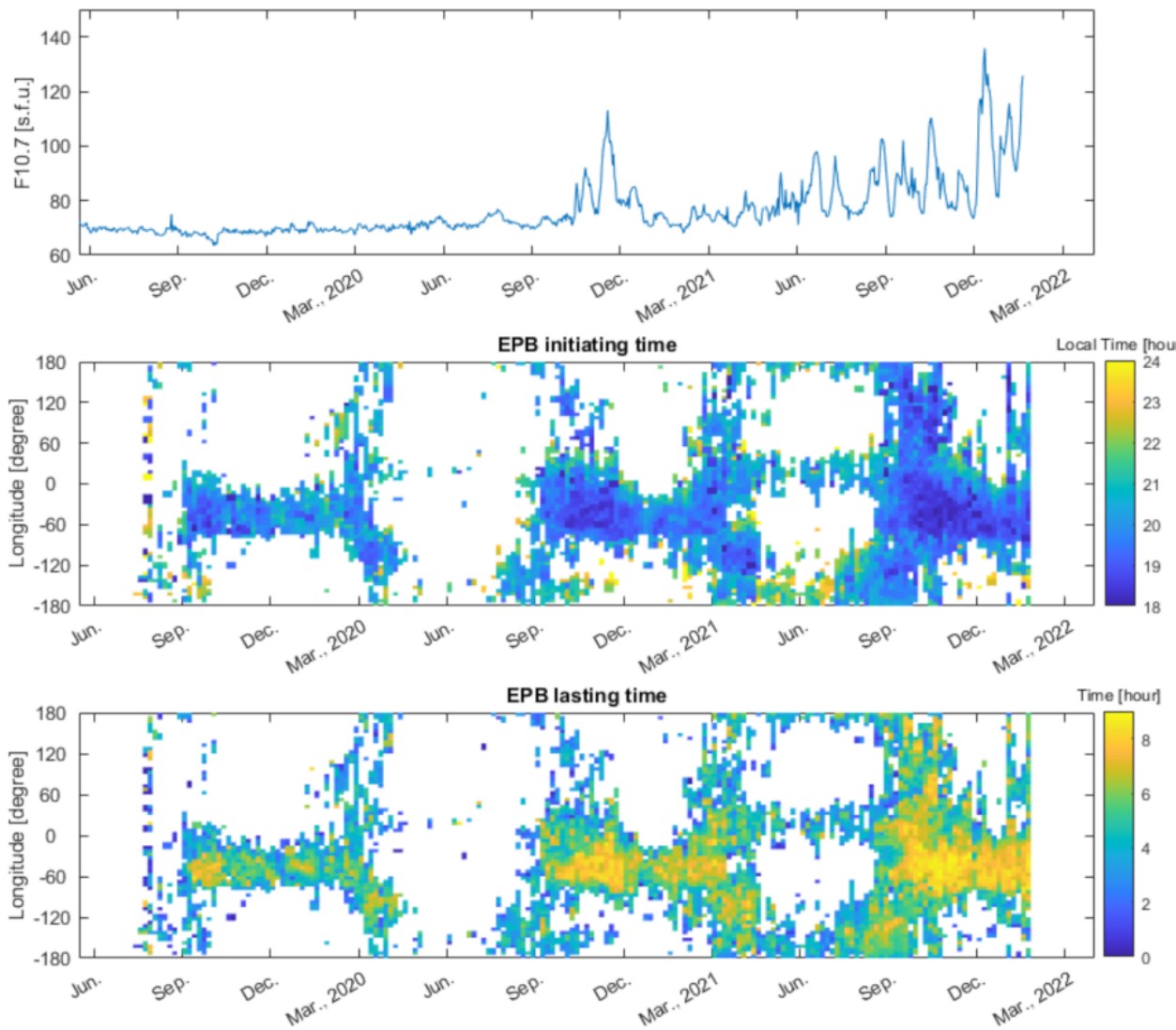

**Figure 7.** The seasonal and annual variation of the initiation and the duration of the F7/C2 S4 EPB detections during August 2019 to February 2022.

The longitudinal and seasonal variations of S4 in Figure 7 agree with those reported by previous in situ measurements. Note that earlier ground-based scintillation measurements showed similarity in the occurrence patterns of stronger S4 with the EPB observations by the Defense Meteorological Satellite Program (DMSP) [109] and Communication/Navigation Outage Forecast System (C/NOFS) [110], suggesting the correspondence between stronger S4 regions and ionospheric irregularities. The solar cycle and seasonal distribution of EPBs revealed in Figure 7 more-or-less agree with those reported in DMSP observations [111–113]. The patterns also show a good agreement with the results from using C/NFOS measurements [114]. Chen et al. [70] noted that the longitudinal variation revealed by F7/C2 S4 better agrees with that of the EPB occurrence in C/NFOS measurements than in DMSP in solar minimum conditions owing to the higher orbit altitude of the latter. However, the pattern may be more identical in higher solar activity conditions where bubbles often grow to higher altitudes. Moreover, there is also good agreement in the S4 pattern in Figure 7 with the seasonal and longitudinal patterns of irregularity distributions reported by Chou et al. [115] by using F3/C measurements.

## 5. Ionospheric Space Weather Forecast

Space weather centers have been set up in many countries to monitor and forecast space weather changes. Global uniform observations of F3/C and F7/C2 ionospheric RO profiles provide the best chance to observe 3D structures of electron density, which for the first time make the ionospheric weather monitoring, nowcast, and forecast possible. The NOAA/SWPC and CWB/SWOO have been assimilating the F3/C and F7/C2 RO ionosphere profiles for space weather monitoring, nowcast, and forecast. A brief introduction of these activities is provided here.

### 5.1. NOAA/SWPC

At NOAA SWPC, the F7/C2 data are being ingested into the GloTEC. GloTEC is a novel global-scale three-dimensional electron density data assimilation scheme that makes use of a Gauss-Markov Kalman filter to optimally estimate ionospheric conditions. The software package uses modern programming languages, practices, and data ingest systems to facilitate near-real-time, as well as retrospective, electron density assimilation, TEC product generation, and dissemination. The model uses ground-based GNSS observations from hundreds of dual-frequency receivers streamed from real-time providers around the world as well as RO observations of slant TEC from the F7/C2 mission. The assimilation output from GloTEC is used to generate maps of vertical TEC, peak F-region density (NmF2), the height of NmF2 (hmF2), and TEC anomaly. The anomaly map is used to identify abnormal ionospheric conditions, such as large TEC enhancements or depletions that affect radio frequency signal propagation. The GloTEC version with only ground-based GNSS data has been in operation since late 2019 to support SWPC's TEC requirement identified by the International Civil Aviation Organization (ICAO). The latest version that includes the F7/C2 data went into operation in March 2022 to support the activities at the Space Weather Forecast Office in SWPC.

Figure 8 shows the GloTEC TEC results with both F7/C2 or ground-based data (top), with only ground-based data (middle), and with only F7/C2 data (bottom) during a minor storm at 10:55 UT on 3 February. At this particular UT time, there was a moderate geomagnetic storm with strong southward Bz (~15) observed in the solar wind measurements. The numbers of measurements for each case distributed globally are also included in the figure. The global ray plots show the increasing ocean coverage when F7/C2 data are utilized. This result also suggests that in this case, the TEC calculation based on only F7/C2 was slightly larger than the TEC derived based only on the ground-based observations.

Figure 9a shows the TEC difference from GloTEC with F7/C2 data and without F7/C2 data at 10:55 UT on 3 February. The ground-based GNSS data were included in both runs. Significant positive bias in the ocean region clearly demonstrated the influence of F7/C2 data in correcting the overall bias in the GloTEC algorithm. Accumulating the comparison throughout the day on 3 February, Figure 9b shows the overall positive impact caused by F7/C2 data in GloTEC at regions where there were five or more observations of the F-region combining both F7/C2 and ground-based data.

Combining ground-based and space-based GNSS signals significantly improved the global coverage for the GloTEC technique and provided us with a better real-time specification of the ionosphere to improve our current products and services. In the near future, SWPC will be obtaining the real-time scintillation data and geolocated measurements from F7/C2. Combining the rate of TEC index (ROTI) and real-time scintillation maps created from F7/C2 measurements, it is expected to largely enhance SWPC's nowcast capability in ionospheric scintillation.

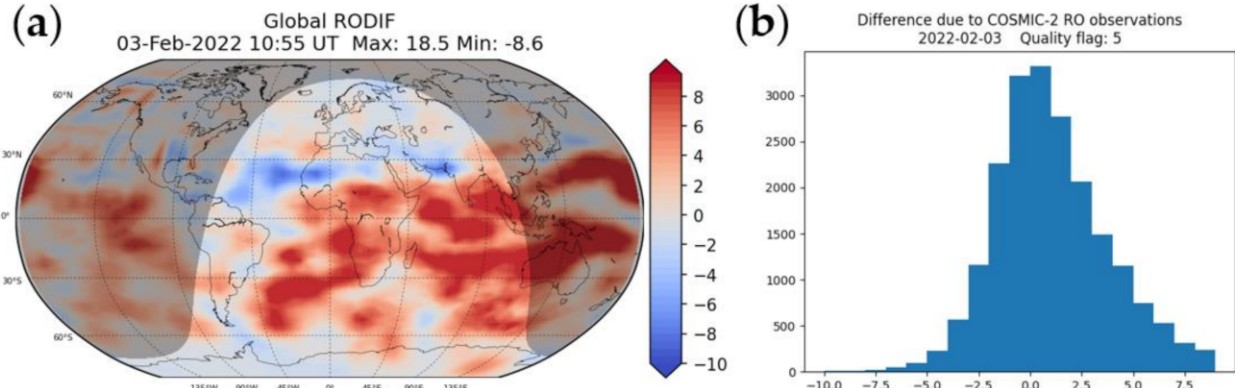

**Figure 8.** GloTEC TEC in units of $10^{16}\text{m}^{-2}$ for 3$^{\text{rd}}$ February 2022 at 1055 UT (**left**) and numbers of measurements (**right**) calculated based on both F7/C2 and ground-based receivers (**top**), only ground-based data (**middle**), and only F7/C2 data (**bottom**).

**Figure 9.** (**a**) TEC difference provided by GloTEC with and without F7/C2 data at 10:55 UT on 3 February 2022. (**b**) Histogram of TEC bias (TECu in the x-axis) calculated from GloTEC with COSMIC-II RO data included.

### 5.2. CWB/SWOO

CWB/SWOO has been creating products for ionospheric weather monitoring and forecast based on the F7/C2 occultations that are routinely published on its website "https://swoo.cwb.gov.tw/V2/page/EN/index.html" (accessed on 1 May 2022). Besides the RO profiles and global scintillation map, global NmF2 and hmF2 of the ionosphere are reconstructed by accumulating F7/C2 observations within an hour period. These products are valuable to monitor the ionospheric condition in near-real-time. Moreover, CWB/SWOO has operated an ionospheric forecast system by additionally assimilating F7/C2 ionospheric RO EDPs since 2020 and produces 6 h ionospheric forecasting hourly "https://swoo.cwb.gov.tw/V2/page/EN/Forecast/IonoForecast.html" (accessed on 1 May 2022). Lee et al. [65] addressed that the F7/C2 RO profiles could adjust the vertical structure of ionospheric electron density closer to reality mainly in the equatorial and low-latitude regions. For instance, Figure 10 displays a set of products, including the RO scintillation map of the L1 and L2 band, and the forecasts of TEC, NmF2, hmF2, and foF2 at 18:00 UT on 28 October 2021 after an ×1.0 solar flare occurred. With intense RO measurements, it could provide global ionospheric features and forecast after particular space weather events promptly as a reference for communication, positioning, and other possible usages.

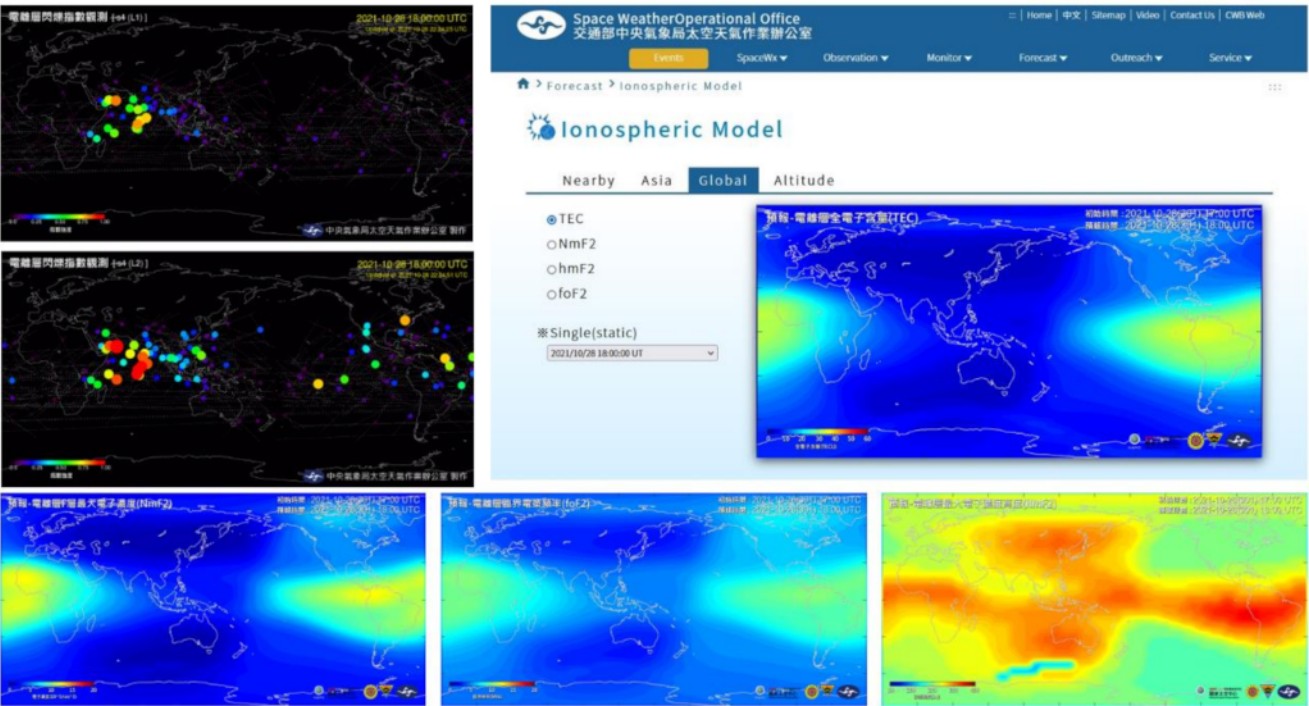

**Figure 10.** The global ionospheric weather of the S4 scintillation map of the L1 and L2 band, and the forecasts of TEC, NmF2, foF2, and hmF2 at 18:00UT on 28 October of 2021. (https://swoo.cwb.gov.tw/V2/page/EN/Forecast/IonoForecast.html) (accessed on 28 October 2021).

### 6. Discussion and Summary

One of the first illustrations of the diurnal evolution of the EIA crests purely from measurement was available to the ionospheric community after Lin et al. [38] showed the 3D global EIA features by using F3/C observations. The results described in the above sections demonstrated that the F7/C2 measurements, with the denser low-latitude soundings (Figure 1), continue such global ionospheric monitoring and provide further opportunities to explore the dynamic interactions in the low-latitude region by adopting similar illustrations used in the previous F3/C studies. The ability to construct monthly global maps of electron density distribution as well the altitude–latitude profiles (movies in

the Supplementary Materials) with higher spatial and temporal resolution windows allows scientists to investigate the longitudinal and seasonal characteristics in greater detail.

The monthly EIA maps (Movie S1) showed distinct seasonal transitions, with a more pronounced electron density occurring in October and the smallest value in June–July. While the electron density variations are influenced by the increasing solar activity, these observations suggest that further investigations and modeling efforts are required to understand such equinox behavior. The strong hemisphere asymmetry seen in September (Figure 3), where rather uniform EIA crests are expected, also showed that denser global soundings are necessary to uncover more detailed electron density variations compared to a rather average picture and to understand the possible role of varying declination in adding to the asymmetry. Similarly, investigations of the global electron density maps at different local times provided further characteristics of the wavenumber-3 and wavenumber-4 patterns (Figure 5) in different months and helped to better distinguish the PDB structures from the longitudinal wave modulations. The unique GNSS RO measurements thus offer exceptional opportunities to further uncover various driving factors that influence and modulate the EIA crests, unravel new features of plasma caves, plasma depletion bays, etc., ushering a new era of ionospheric monitoring through nowcast and forecast models, and global scintillation maps for space weather.

Despite such advantages from the nearly 5000 RO soundings per day (Figure 1), the F7/C2 observations are still inadequate to generate daily global electron density maps, requiring combining several days of measurements. This comes at the expense of valuable information about the day-to-day variability of the ionosphere. It has been shown that the day-to-day variability accounts for about 10–30% of electron density variation during daytime and may be about 35–50% at night [116,117]. Modeling studies have showed that about half of such day-to-day variability could be of meteorological origin [118,119], and geomagnetic activity plays a dominant role in imparting such daily variations [120]. Thus, it is important to have daily global observations, and the current RO measurements are insufficient. However, the active data assimilation efforts by using F7/C2 observations greatly help to overcome this limitation of the RO observations.

The space weather forecasting model used at CWB/SWOO assimilates the RO measurements to provide daily global electron density maps, with a 6 h forecast capability [65]. By assimilating more observations, realistic global electron density maps can become available to examine physics that dominate electron density variations. At NOAA/SWPC, the GloTEC that are generated by assimilating both F7/C2 RO measurements and GNSS TEC have become a useful tool to support the alerts and warnings issued by space weather forecasters. In addition to such dedicated centers offering daily ionosphere maps, there are also ongoing efforts to assimilate the F7/C2 RO measurements to improve the performance of physics-based models [121,122]. Another major source of daily 3D electron density profiles is the GIS electron density constructed by assimilating F7/C2 RO and ground-based GNSS slant TECs [61,62,64]. GIS provides global electron density profiles at every hour with 20 km altitude resolution and $2.5° \times 5°$ in latitude and longitude spacing. The GIS electron density has been used to examine ionosphere variations during magnetic storms [69] and electron density modulations during SSW [68], and to examine day-to-day ionosphere variability [67].

In summary, the GNSS RO soundings have revolutionized ionospheric research ever since the launch of F3/C satellites. The current F7/C2 mission, with even denser observations at the low-latitude region, offers more detailed electron density variations and near-real-time plasma irregularity monitoring capability to open up numerous opportunities in advancing space weather research. Furthermore, the F7/C2 measurements greatly contribute to constructing daily 3D global electron density maps to provide realistic and near-real-time electron density distribution. The additional advantages of the denser F7/C2 samplings demonstrated here shows that extending similar measurements to cover mid- to high-latitude regions in the future and also allowing for extended altitude coverage would extremely benefit global ionospheric investigations and help to further advance the

ongoing modeling and forecast efforts. In conclusion, the GNSS RO observations offer a promising era for ionosphere space weather, especially as we are approaching another solar maximum.

**Supplementary Materials:** The following supporting information can be downloaded at: https://www.mdpi.com/article/10.3390/atmos13060858/s1. Movie S1: Movie showing the 3D electron density structures similar to Figure 2, but for each month at each UT hour; Movie S2: Movie showing local-time maps of EIA evolution similar to Figure 3, but for each month.

**Author Contributions:** Conceptualization, J.-Y.L.; J.-Y.L., C.-H.L., P.K.R., C.-Y.L., F.-Y.C., I.-T.L., T.-W.F., D.F.-R., S.-P.C. equally contribute to all other aspects. All authors have read and agreed to the published version of the manuscript.

**Funding:** This study was supported by the Taiwan Ministry of Science and Technology grant MOST 108-2119-M-008-001. This work was financially supported by the Center for Astronautical Physics and Engineering (CAPE) from the Featured Area Research Center program within the framework of Higher Education Sprout Project by the Ministry of Education (MOE) in Taiwan.

**Data Availability Statement:** The FORMOSAT-3/COSMIC (F3/C) and FORMOSAST-7/COSMIC-2 (F7/C2) data used in this study were retrieved from Taiwan Analysis Center for COSMIC (TACC, https://tacc.cwb.gov.tw/v2/download.html, accessed on 1 May 2022).

**Acknowledgments:** The authors acknowledge the Taiwan Ministry of Science and Technology and the Center for Astronautical Physics and Engineering (CAPE) supported by the Ministry of Education (MOE) of Taiwan for supporting this study. The authors also acknowledge the Taiwan Analysis Center for COSMIC (TACC) for the FORMOSAT-3/COSMIC (F3/C) and FORMOSAST-7/COSMIC-2 (F7/C2) data used in this study.

**Conflicts of Interest:** The authors declare no conflict of interest.

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
