# Peer review of "Advances in Ionospheric Space Weather by Using FORMOSAT-7/COSMIC-2 GNSS Radio Occultations"

_atmosphere, doi:10.3390/atmos13060858_

Round 1

Reviewer 1 Report

Dear Editor,

The paper by Liu et al. titled “Advances in Ionospheric Space weather by Using the GNSS radio occultation” provides an overview of development GNSS observation by COSMIC 1 to COSMIC 2 mission.    Both missions have achieved or is reaching their potential for advancing the ionospheric research and space weather potential.   The paper is very well written and should be considered for publication.   I have only some minor comments.

  1. It is not clear the ionospheric density data shown in Figure 2 are model simulation results or re-gridded density profile data.
  2. Figure 7 shows longitudinal and seasonal variation of the S4 EPB results. The results are similar to some of the earlier observations.   It will be interested to provide some discussion of the results in comparison with earlier observations from DMSP or CNOFS.
  3. In discussion and summary, the authors showed a great regret that the polar orbiting portion of the COSMIC 2 was not realized resulting in lack of high latitude observations. Author should mention that many commercial GNSS RO missions are current operating, although the data are not widely available now.   We should encourage them to distribute more non-real time data to the research community to complement COSMIC 2 data set. 
  4. It should also be pointed that COSMIC 2 data are higher quality than most of the commercial data products and can serve the source for calibration.

Reviewer 2 Report

Title: Advances in Ionospheric Space Weather by Using GNSS Radio Occultation

The manuscript has serious flaws regarding concepts about ionospheric RO. A proper understanding of the technique is required for writing a proper overview paper. The text is badly written in many parts and a Figure is missing. The authors have even used a Wikipedia text.

In addition, the literature review is still incomplete. In details, it is only reviewed EIA electron densities, ionospheric wave numbers and plasma bubble detections. The reviewer suggests the authors to make a much larger literature review to see all the applications that ionospheric RO allows when studying the ionosphere between -35 to +35 degrees.

I cannot accept the paper in the way it is presented now.

“Atmospheric RO relies on the detection of the changes in a radio signal as it passes through a planet's atmosphere, i.e., when it is occulted by the atmosphere.”
The whole sentence comes directly from Wikipedia. And the reviewer does not think this an accurate statement.

“demonstrated active limb sounding of the Earth’s atmosphere and ionosphere”
Sentence retrieved from doi: 10.1186/BF03352376

“unaffected by measurement location or observing conditions [4]”
Ionospheric RO accuracy is affected by both.

“Interested readers may refer to Anthes et al. (2008) [5]”
wrong format.

“enabling the researchers worldwide with the opportunity to examine 3D ionospheric characteristics in greater detail, unlike what was possible before.”
It was possible to examine the 3D ionosphere in great detail before GPS-RO.

“The following years witnessed a large influx of quality research articles appearing in international journals, making use of the rare ionospheric sounding measurements from a unique space-based observing platform, portraying 3D electron density features and their temporal evolutions for the first time from actual measurements.”
There were RO measurements before 2006.

“Liu et al. (2022) [6] provide”
Wrong format. This occurs several times. I will stop addressing here the remaining ones.

“leading to the identification of new features such as plasma caves”
Plasma caves are not an feature of the ionosphere. It appears due to the inaccuracy of the RO at certain locations.

“This paper attempts to extend the work of Liu et al. (2022) [6]”
The reviewer cannot find the reference paper since it is not yet published. Therefore, it is not possible to check the main differences between both manuscripts, as the authors are requiring the readers to do.

“manifesting as two enhanced peaks (crests) of electron density surrounding a narrow belt of low values of ionization (trough)”
The electron density at the Equator is not low

I could not find the supplementary material.

“Chen et al. (2021) [71] further improves the F3CGS4 model by using the whole S4-index profile data and constructs the first global probability model for the S4-index of L-band scintillations.”
There existed probability models for the S4-index before 2021

Figure 9 does not exists, despite the caption is there.

Reviewer 3 Report

This paper has done lots of work in investigating the new achievements in ionospheric fields by using COSMIC2 ionospheric products. However, the paper is a little hard to follow since the writing is less-well organized. Authors should re-organize the paper, highlights the outcomes/new-findings of their research. My comments are as below:

Major comments:

1) It seems that authors mainly use COSMIC-2 data, can the title highlights using of COSMIC-2 ionospheric products?

2) The abstract of this paper spends most of the paragraph in introducing the background of this paper, and only use one sentence to introduce the purpose of this paper. However, an abstract should briefly summarize the methodology used and the new findings of the paper.

3) At the end of both sections 1 and 2, there is an introduction of the aim of this paper, which is redundant. Please combine them and maybe put them at the end of Section 1.

4) This paper lacks a comprehensive literature review on current achievements on ionospheric findings using COSMIC data. Section 1 spends most of the paragraph in introducing COSMIC mission, however, the focus of this paper should put on ionospheric findings. Some literature review is available in the results section. However, they should be put in the introduction section.

5) Some abbreviations such as LT should be introduced.

6) Authors should clearly summarize the new findings of this paper. The findings should be more in a logical sequences other than in a discrete way.

Detail comment:

Line 32: “Atmospheric RO” is not very often used, often used phase is GNSS/GPS RO.

Line2 35, there is a space at the beginning of this line should be avoided

Line 112, “The focus is to give the readers a glimpse of the” should be changed to “the focus of this paper is to provide an over view…”

Line 188: “This section will overview” should be changed to “This section provides an overview on”

Line 255: Figure 2 caption: “1200 UT” should be changed to “12:00 UTC time”

Round 2

Reviewer 3 Report

This paper has followed my suggestions.

Minor comment:

It is inappropriate for reference [81] to be listed in the first citation of the paper. Please adjust the citation order of references. 
